# Clinical Application of Induced Hepatocyte-like Cells Produced from Mesenchymal Stromal Cells: A Literature Review

**DOI:** 10.3390/cells11131998

**Published:** 2022-06-22

**Authors:** Yanina Bogliotti, Mark Vander Roest, Aras N. Mattis, Robert G. Gish, Gary Peltz, Robin Anwyl, Salah Kivlighn, Eric R. Schuur

**Affiliations:** 1Hepatx Corporation, Palo Alto, CA 94304, USA; ybogliotti@hepatx.com (Y.B.); mvanderroest@hepatx.com (M.V.R.); ranwyl@hepatx.com (R.A.); skivlighn@hepatx.com (S.K.); 2Department of Pathology, School of Medicine, University of California, San Francisco, CA 94143, USA; aras.mattis@ucsf.edu; 3Robert G. Gish Consultants, LLC, San Diego, CA 92037, USA; rgish@robertgish.com; 4Department of Anesthesia, Pain and Perioperative Medicine, Stanford University School of Medicine, Stanford, CA 94305, USA; gpeltz@stanford.edu

**Keywords:** liver, hepatocyte, mesenchymal, stem cell, cirrhosis, therapy, differentiation

## Abstract

Liver disease is a leading cause of mortality worldwide, resulting in 1.3 million deaths annually. The vast majority of liver disease is caused by metabolic disease (i.e., NASH) and alcohol-induced hepatitis, and to a lesser extent by acute and chronic viral infection. Furthermore, multiple insults to the liver is becoming common due to the prevalence of metabolic and alcohol-related liver diseases. Despite this rising prevalence of liver disease, there are few treatment options: there are treatments for viral hepatitis C and there is vaccination for hepatitis B. Aside from the management of metabolic syndrome, no direct liver therapy has shown clinical efficacy for metabolic liver disease, there is very little for acute alcohol-induced liver disease, and liver transplantation remains the only effective treatment for late-stage liver disease. Traditional pharmacologic interventions have failed to appreciably impact the pathophysiology of alcohol-related liver disease or end-stage liver disease. The difficulties associated with developing liver-specific therapies result from three factors that are common to late-stage liver disease arising from any cause: hepatocyte injury, inflammation, and aberrant tissue healing. Hepatocyte injury results in tissue damage with inflammation, which sensitizes the liver to additional hepatocyte injury and stimulates hepatic stellate cells and aberrant tissue healing responses. In the setting of chronic liver insults, there is progressive scarring, the loss of hepatocyte function, and hemodynamic dysregulation. Regenerative strategies using hepatocyte-like cells that are manufactured from mesenchymal stromal cells may be able to correct this pathophysiology through multiple mechanisms of action. Preclinical studies support their effectiveness and recent clinical studies suggest that cell replacement therapy can be safe and effective in patients with liver disease for whom there is no other option.

## 1. Introduction

### 1.1. Treatment of Acute and Chronic Liver Disease Is a Major Unmet Global Need

Liver disease is the 11th leading cause of death worldwide [1]. The 2017 Global Burden of Disease study estimated more than 1.3 million deaths, or 2.4% of all deaths, were due to liver disease. The estimated direct cost of liver disease is greater than USD 20 billion in the United States alone [2]. The causes of liver disease range from rare genetic disorders, causing loss of a critical gene product, to fulminant liver failure and a catastrophic loss of liver function due to progressive liver damage, which ultimately leads to decompensated liver failure and cirrhosis. Cirrhosis accounts for the largest percentage of mortality from liver disease [3]. There are multiple etiologies for cirrhosis, including alcohol-associated liver disease, metabolism-associated fatty liver disease, and viral infection including viral hepatitis B, C, and/or D. Regardless of the etiology of the disease, patients experience similar patterns of liver dysfunction in later stages of the disease. Currently, only liver transplant is curative for this late-stage disease. Here, recent results of studies employing novel cell therapies using hepatocyte-like cells manufactured from mesenchymal stromal cells to treat liver disease are reviewed. These regenerative strategies show promise for both acute and chronic use in addressing the complex nature of late-stage liver disease.

### 1.2. Pathophysiology of Liver Disease Is Very Complex

#### 1.2.1. Three Basic Pathological Processes Contribute to the Development of Cirrhosis

Although different disease-causing agents affect the liver through distinct molecular mechanisms, the resulting pathophysiology converges on common processes and pathways that ultimately result in cirrhosis (Figure 1). Hepatocyte injury (from alcohol or metabolic imbalances, viral injury and inflammation, and genetic or autoimmune cholestatic injury) all result in cellular death or programed cellular death that triggers inflammation followed by regenerative tissue repair processes to replace the damaged cells [4]. Chronic injury starts out as acute, but with consistent insults converts into chronic inflammation and exhibits an aberrant healing response that leads to a dysfunctional tissue architecture [4,5]. Chronic inflammation leading to and combined with an increasingly distorted tissue architecture ultimately leads to the late sequelae of cirrhosis: portal hypertension, hemodynamic issues, and the loss of hepatocyte function. Patients ultimately succumb to the consequences of these, which include gastrointestinal bleeding, spontaneous bacterial peritonitis, hepatorenal syndrome, hepatic encephalitis, and cardiac-related dysfunction [4].

#### 1.2.2. Hepatocyte Injury Is the First Step

These three common themes, hepatocyte injury, inflammation, and aberrant tissue repair, have been characterized in detail (Figure 1). The initial effect of the injurious agent is to induce stress in hepatocytes, which ultimately leads to hepatocyte death. In hepatitis B and C viral infections, viral replication in hepatocytes results in cellular stress with limited hepatocyte apoptosis which, in turn, stimulates a T cell-mediated killing of the virus-infected cells [6]. Acetaminophen overdose, as an example of chemical injury, directly causes hepatocyte death due to an accumulation of N-acetyl-p-benzoquinone imine (NAPQI), a highly reactive and toxic byproduct of cellular metabolism, resulting in the exhaustion of reducing agents and hepatocyte necrosis [7]. Excessive alcohol consumption initially induces triglyceride formation, leading to steatosis. Continued excessive intake results in the intracellular generation of toxic compounds via the cytochrome system and reactive oxygen species that result in hepatocyte death [8]. Hepatocyte death, independent of the initial insult, causes the release of damage-associated molecular patterns (DAMPs), which activate the innate immune system in the liver and systemically lead to inflammation.

#### 1.2.3. Inflammation and Cytokine Secretion Contributes to the Cascade of Liver Injury

The second theme is inflammation. Terminal hepatocyte injury results in at least two initiating factors: the release of DAMPs and other molecules that trigger the immune system, and the release of cytokines that further amplify the immune response, sensitize hepatocytes to further death signals, and lead to systemic hemodynamic dysfunction [8,9,10]. The liver is the intermediary between the gut-derived blood supply, with its wide variety of substances from the gut, and systemic circulation. In this role, the liver is positioned to guard against infectious agents and endotoxins and to detoxify injurious compounds. The liver contains many macrophages, termed Kupffer cells. These specialized cells respond to DAMPs and cytokines secreted by T cells and other cells (sinusoidal endothelial cells and hepatic stellate cells) during the damage phase by secreting additional cytokines (interleukin [IL]-1 beta, IL-6, tumor necrosis factor alpha [TNF alpha] alpha, and others) to amplify the immune response by attracting neutrophils and other immune cells and further sensitizing hepatocytes to environmental cues [8,9]. In addition, in the case of alcohol consumption, pathogen-associated molecular patterns and endotoxins are also involved due to the leakage of Gram-negative bacteria, endotoxins, and cytolysin into the portal system, further amplifying inflammation.

#### 1.2.4. Aberrant Tissue Repair Leads to Fibrosis

The third element to the liver’s response to chronic injury is fibrosis. Normally, tissue damage stimulates a complex tissue repair process. The initial response includes an inflammatory phase during which secreted cytokines attract inflammatory macrophages and neutrophils, resulting in the phagocytosis of cellular debris and the secretion of additional cytokines responsible for stimulating the resolution of the injury. Recruitment of fibroblasts, endothelial cells, and other resident cells, such as hepatic stellate cells, leads to the neovascularization and remodeling of the extracellular matrix (ECM) as a tissue repair progresses. The cytokine most responsible for promoting fibrosis is transforming growth factor beta (TGF beta), while the balance of matrix metalloproteinases and their inhibitors, the tissue inhibitors of metalloproteinase (TIMPs), can either contribute to fibrosis or promote its resolution [11]. Hepatic stellate cells, which are normally quiescent and reside between the sinusoidal endothelial cells and hepatocytes where they serve as a reservoir of vitamin A, are key cellular mediators of fibrosis. Under conditions of tissue injury and inflammation, stellate cells are activated, primarily by TGF beta 1 from T cells, and begin to secrete ECM and various cytokines as well as differentiate to myofibroblasts with an ECM secretory phenotype [5,11]. Under chronic stress or long-term ongoing tissue injury, the above processes continue without appropriate tissue damage resolution and with excess collagen matrix deposition, which ultimately leads to the scarring observed in cirrhosis.

At the point at which scarring of the liver reduces its capacity to meet the physiologic needs of the body, hemodynamic, immunological, and biochemical dysfunction are observed [4]. The scarred tissue of the cirrhotic liver leads to portal hypertension with its sequelae of ascites, gastrointestinal bleeding (especially esophageal varices), hepatic encephalopathy, coagulopathy, and kidney dysfunction. As a result of the liver now shunting blood, a lack of the usual dynamic blood return to the heart can result in inadequate return and additional cardiac stress. Loss of hepatocyte functional capacity contributes further to the disease through reduced hemostasis capacity and reduced metabolite detoxification (especially the increased accumulation of toxins, including systemic ammonia). The resulting immunologic deficit contributes to the presence of gut microbes in the systemic circulation and associated sepsis, as well as further sustained inflammation. While patients can live with compensated cirrhosis for quite some time, the interplay of these late-stage factors presents as the classic decompensated cirrhosis symptoms of ascites, bleeding esophageal varices, hepatic encephalopathy, bacterial peritonitis, and hepatorenal syndrome.

This self-reinforcing cycle of hepatocyte damage followed by inflammation, followed by more hepatocyte damage, leading to hepatocyte death, and backfilling with connective tissue is the common theme that underlies progressive loss of liver function. While the complete resolution of the disease process requires multiple targets of therapy, the modulation of inflammation and the replacement of hepatocyte function, either directly or by trophic support of stressed hepatocytes, will likely be key. Achieving this will have a significant clinical impact by moving patient physiology toward normal homeostasis and forestalling critical clinical events, such as multi-organ system failure.

### 1.3. Cell Therapy Has the Potential to Address the Liver Disease Pathology

Currently, treatment of liver disease in many cases consists of supportive care for target symptoms. For example, a variety of drug strategies have been tested for their ability to either quell inflammation or provide trophic support for liver tissue in patients with various forms of liver disease. Steroids are a first-line therapy for hepatitis, even though they have little or no impact on survival [12]. Data demonstrating the centrality of IL-1 in the inflammatory process and the natural anti-inflammatory effect of interleukin 1 receptor antagonist (IL-1RA) led to its development as an anti-inflammatory therapy (marketed as anakinra) [13]. Similarly, the pharmacologic inhibition of fibrosis has been tested to block the development of cirrhosis. Strategies include peroxisome proliferator-activated receptor gamma inhibitors, which block hepatocyte apoptosis, and the concomitant stimulation of fibrosis and the inhibition of hepatic triglyceride formation by inhibiting the farnesoid X receptor [5,11]. To date, these single-point interventions have not significantly impacted clinical disease. Once the disease reaches the decompensated state, no therapy—other than orthotopic liver transplantation—has been successful. Due to a chronic shortage of donor organs, this therapy is not viable for most patients.

Because of the often-intractable nature of liver disease, there has been a great deal of interest in the application of cell therapies to liver disease based on their potential to reverse the ongoing pathophysiology of liver disease using novel mechanisms of action [13,14,15]. Hepatocyte transplantation and engraftment has been demonstrated in rodent models of liver disease [16,17]. This regenerative strategy has the potential to deliver the benefits of organ transplantation in the long term. However, cell therapy using hepatocyte-like cells derived from mesenchymal stromal cells (mHeps) has the potential to orchestrate multiple mechanisms of action, thereby enabling novel anti-inflammatory and trophic strategies that may be useful in the context of the complex pathophysiology of liver failure. A large body of literature has demonstrated the ability of mesenchymal stromal cells (MSC) to deliver multiple beneficial effects in vivo, including the suppression of inflammation and the stimulation of tissue regeneration [13,15,18]. The multipotent nature of these cells also suggested that they could be potentially useful in cell replacement strategies, which have been tested in multiple studies [19,20]. Importantly, the differentiation of MSC-type cells to mHeps can preserve some of the desirable features of MSCs while adding useful hepatocyte functions, thus creating cell-based therapeutics that can address the complex physiology of liver failure.

We review, below, recent studies employing mHeps derived from MSCs to treat liver disease (Table 1). The abundant supply of MSCs as raw material, the ability to reliably manufacture mHeps, and their potential to orchestrate complex therapeutic effects make this an attractive strategy for generating cell therapies for liver disease. The potential for successful clinical translation is assessed below.

## 2. MSC-Derived Hepatocytes Show Benefit as Liver Therapies

### 2.1. MSCs Can Be Directed into the Hepatocyte Lineage

MSCs were first described in bone marrow by Friedenstein in 1970, and they were subsequently characterized by Pittenger et al. as self-renewing cells that were capable of differentiation to adipocytes, osteoblasts, and chondrocytes, which illustrates their multipotent nature [15]. Subsequent studies demonstrated that these cells are resident in most or all adult tissues. Most studies have used MSCs from bone marrow; however, adipose tissue and Wharton’s jelly are rich sources of these cells as well.

Isolation of these cells is straightforward. They are defined by their ability to adhere to common plastic tissue culture surfaces following dissociation of the tissue by enzymatic treatment [29,30]. There is an extensive amount of research in the literature on the production of MSCs for experimental and clinical use [31,32,33,34].

### 2.2. Evolution of mHep Differentiation from MSCs: In Vitro Characteristics to Therapeutic Utility

Because of the multipotent nature of MSCs, there was enthusiasm for using this plentiful, easy-to-handle cell source as a basis for hepatocyte transplantation using MSC-derived hepatocytes. An initial demonstration of MSC-like cells’ ability to undergo differentiation to hepatocytes was reported by Schwartz et al. [20]. They reported the differentiation of multipotent adult progenitor cells isolated from bone marrow using a single serum-free medium containing a combination of fibroblast growth factor (FGF), hepatocyte growth factor (HGF), insulin transferrin selenium solution (ITS), and dexamethasone in a monolayer culture on a collagen and Matrigel-coated plastic surface [20]. The resulting cells expressed morphologic, phenotypic, and functional characteristics of hepatocytes after 14–28 days of treatment. However, the differentiation was not uniform nor efficient (Table 1).

Evolution of the mHep differentiation procedure included several variations on sequential exposure to cytokines and differentiation-inducing factors (Table 1). Snykers et al. reported the sequential exposure of bone marrow MSCs growing on plastic coated with collagen type I to FGF4, then by HGF, and subsequently to a mixture of HGF, ITS, and dexamethasone [23]. The resulting cultures contained more than 85% mHeps, defined by the expression of hepatic markers, including cytochrome P450 gene products. Additional combinations of cytokines added in sequence or as a mixture that have been successful for the differentiation of adherent MSCs include FGF, HGF, and oncostatin M. Dexamethasone, ITS, and nicotinamide have been observed to have a synergistic effect on differentiation in these systems. While this improved the efficiency of mHep production, it was at the cost of added complexity and did not accelerate the process.

Banas et al. used MSCs from adipose tissue (ASCs), an abundant source of MSCs, and demonstrated that these cells could be differentiated to mHeps quickly (13+ days) by first differentiating them to endoderm-like cells followed by hepatic differentiation [26,28]. Xu et al. extended this finding by showing that mHeps from ASCs can be rapidly produced at scale using a simplified endoderm-hepatic protocol in suspension, thus clearing many of the hurdles to the practical production of mHeps for clinical use [19].

Both Camussi and Sokal independently employed liver-derived MSCs for generating mHeps [35,36,37,38]. Both groups reported a procedure for the isolation of stem cells from human liver fragments that are capable of extended proliferation in vitro. The Camussi group produced stem cells that possess several of the phenotypic (CD44+, CD90+, CD73+, CD31−, CD45−) and functional (trilineage differentiation) properties of MSCs and respond to hepatic differentiation protocols, such as those described above. These cells express hepatocyte markers and functionality, including albumin (ALB), cytokeratin 18 (KRT18), HGF, and cytochrome proteins. Cells from the Sokal group have a similar phenotype; however, they do not undergo trilineage differentiation, nor do they express KRT18 [38].

As can be seen from the discussion above, various differentiation protocols have been applied to MSCs from several different sources (bone marrow, adipose tissue, and liver) and can be directed into the hepatocyte lineage. Based on the published results, mHeps from each of these sources (as well as umbilical cord and other tissues) have similar properties in vitro and similar effects in preclinical models as discussed in the section that follows. Therefore, no definitive conclusion can be drawn at this time regarding the relative biological performance of MSCs by source. The choice of protocol and MSC source may be driven more by other factors, such as abundance and simplicity.

### 2.3. Therapeutic Use of mHeps

Therapeutic utility of mHeps has been demonstrated in a variety of different types of liver disease (Table 2). Several studies have demonstrated the engraftment of these mHeps in rodent models [19,27,28,36,39,40,41,42,43,44,45] and pigs [46]. The efficiency of engraftment has generally not been reported, although Aurich et al. presented data indicating that upwards of 20% of mouse livers were composed of mHeps at 10 weeks [41]. Xu and Peltz inferred that mHeps composed 5–10% of the livers of TK-NOG mice 2 months post-transplantation, based on serum human albumin measurements [19]. These results suggest that mHeps from MSCs can engraft. Once engrafted, the mHeps appear to undergo additional maturation [19,47,48].

The modeling of the therapeutic applications of these cells is summarized in Table 2. In these studies, the carbon tetrachloride (CCl_4_) model of liver injury using a single dose to induce an acute injury in immunocompromised mice is the most often-used model [22,25,26,27,28,43,44,45]. Other models employed TK-NOG mice (immunocompromised mice with the thymidine kinase gene expressed via the albumin promoter [49]) treated with ganciclovir [19], partial hepatectomy [38,39,40,45], galactosamine/lipopolysaccharide [35], galactosamine [50] and acetaminophen toxicity [36]. The routes of administration (ROA) varied and included tail vein, central venous administration, direct implantation in the liver, and intrasplenic (portal vein).

**Table 2 cells-11-01998-t002:** Studies of the effect of MSC-derived mHeps on liver disease and liver disease models.

Starting Cell Type	Type of Differentiation	Length of Differentiation	Phenotypic Characteristics	Functional Characteristics	Model	Outcomes	Reference
MSCs NOS	1-stage hepatic	28 days	Up: KRT18, TO, AAT, HNF4A, Col II, aggrecan, ALB, CYP3A4	ALB secretion, urea synthesis	Rat, partial Hx Direct implantation Harvest at 14 days	Engraftment (IHC) Histology: ALB, human nuclear antigen	Ong 2006 [39]
Bone Marrow MSCs	1-stage hepatic	15 days	Up: KRT18, CX32, Hep par 1, PCK1, CK19, AFP, CX43, CYP3A4, TFN	Glycogen storage, urea synthesis	Pfp/Rag2 mice Partial Hx with propranolol to inhibit hep replication 10^6^ cells intrasplenic at time of partial Hx Harvest at 7 days	Engraftment (periportal, IHC)	Aurich 2007 [40]
ASC, hu	1-stage hepatic	15 days	Up: ALB, PCK, CD26 (PCR); ALB, PCK, CD26 (immunofluorescence)	Albumin secretion	Pfp/Rag2 mice Partial Hx with propranolol to inhibit hep replication 10^6^ cells intrasplenic at time of partial Hx Harvest at 7 days	Engraftment measured by flow cytometry. 21–26% of liver cells positive for hu hepatocyte markers.	Aurich 2009 [41]
Bone Marrow MSCs	1-stage hepatic	14 days	Up: CYP genes (1A1, 3A4), Glycogen storage, Western (PCK, CYP1A1, CYP3A4, GS)	Glycogen storage	Partial hepatectomy Pigs 10^8^ cells IV right after surgery Harvest at 24 h	Decreased: AST, ALT (25–60% reduction), ammonia (60% reduction), lactate; thrombospondin, TGF beta, SMAD signaling No change: INR ICG Gene array measurements in liver and lung tissue Increased: ATIII expression	Nickel 2021 [45]
Liver MSCs	In vivo	None	Up: CD73, CD90, CD44, CD29, CD105 (20%), ALB, AFP, KRT18 (15%), CK8 (11%), VIM, NES Off: CD34, CD45, CD14, CD117, CD133, CK19, ACTA2, NCAM, Stro-1, CYPs	ALB secretion, urea synthesis, CYP activity (after diff with HGF and FGF4)	SCID mice Acetaminophen IP 2 × 10^5^ LSC IV Harvest at 7 or 30 days	Engraftment: HLA I stain	Herrera 2006 [36]
Liver MSCs	In vivo	None	See Herrera 2006 [36]	See Herrera 2006 [36]	SCID mice GalN/LPS IP Treatment 30 min post GalN/LPS 11 treatment groups, incl cells IV, IP, and intrahepatic, conditioned medium Harvest at 7 h and 3 days	Up: survival with LSC and CM, BRdU incorp Down: AST, ALT (30% reduction), NH_4_ (50% reduction), apoptotic nuclei Engraftment	Herrera 2013 [35]
Liver MSCs	In vivo	None	Up: ALB, AFP, VIM, NES, OCT4, Nanog, CK8/18, SSEA4 SOX2, CD29, CD73 Off: ACTA2 Other: telomere length, gene array analysis and comp to BM MSCs	N/D	SCID mice NASH induced by MCDD diet 1.5 × 10^6^ cells at weeks 1, 2, or 3 IV by tail vein Harvest at week 4	Function improved, time-dependent: AST, ALT (reduced 30–50%), ALB, BUN (30% reduction at highest dose) Histology improved, time-dependent: fibrosis, PCNA Gene expression in liver, time-dependent: TGFB1, COL I, ACTA1, IL1B, INFG Gene expression in liver not improved, time-dependent: TNF alpha Function improved dose-dependent: AST (1 dose level), ALT, ALB (1 dose level), BUN (1 dose level) Histology improved, dose-dependent: fibrosis, CD45+ cells Histology not improved, dose-dependent: steatosis Engraftment: pos by alpha sat-ch17	Bruno 2019 [42]
ASC, hu	2-stage endoderm/ hepatic	13 days	Up: EPCAM, FOXA2, SOX17, ALB, ASGR1, Down: CD105	Glycogen storage, LDL uptake, albumin secretion, urea synthesis, CYP3A4 activity	TK-NOG mice Direct implantation 2 × 10^6^ cells Harvest at 2 months	Engraftment, ALB secretion, IHC	Xu 2014 [19]
MSCs NOS	4-stage endoderm/ hepatic	16 days	Hepatocyte morphology, albumin synthesis, urea metabolism, and sequential mRNA expression and protein expression of the hepatocyte markers SOX17, FOXA2, HHEX, GATA4, HNF4A, AFP, ALB, and CK18	Glycogen storage, LDL uptake, CYP activity, ICG uptake and release, albumin secretion	GalN IP Lewis rats 100 spheres Intrasplenic	Survival ALT (40% reduction) Immunohistochemistry for human markers Engraftment	Ramanathan 2015 [50]
ASC, hu	1-stage hepatic	21–28 days	Up: AFP, ALB	LDL uptake, urea synthesis	CCI_4_ acute, IP NOD-SCID mice 1 × 10^6^ cells IV, tail vein Transplant at 48 h post CCI4 Harvest at 3–10 days post-transplant	Engraftment (IF)	Seo 2005 [22]
ASC, hu	2-stage hepatic	35 days	Up: ALB, AFP, TTR, TDO, CYP7A1, HNF4A (41 days)	Glycogen storage, LDL uptake, albumin secretion, ammonia clearance	CCI_4_ acute BALB/c nu-nu IV, tail vein Cells at 24 h, harvest at 48 h	Engraftment (IHC) Histology: ALB, human nuclear antigen detection	Banas 2007 [26]
ASC, hu	1-stage hepatic	31 days	Up: ACTC, PDX-1, SOX1, AAT1, KRT18, CYP1B1, CYP3A4, glutamine synthase	Albumin secretion, Glycogen storage, CYP activity, urea synthesis	CCI_4_ chronic, 12 week NOD-SCID mice Transplant under kidney capsule Harvest at 7 days post-transplant	ALB improved, serum T bil reduced (35%), AAT synthesis, engraftment	Okura 2010 [44]
ASC, hu	3-stage endoderm/ hepatic	28 days	Up: SOX17, CXCR4, AFP, ALB, AAT	ALB secretion, CYP3A4 activity, urea synthesis	Cl_4_ acute BALB/c nu-nu mice Transplanted under kidney capsule Cells at 4 h post CCI_4_ Harvest at 14 days post-transplant	Decreased: ALT, AST (25–40% reduction), T bil (25–70% reduction) Decreased NH_4_, increased urea Engrafted cells by IHC	Saito 2021 [43]
ASC, hu	3-stage endoderm/ hepatic	13 days	Up: ALB, TOD2, FOXA2	Glycogen storage, LDL uptake	CCI_4_ acute BALB/c nu-nu IV, tail vein Cells at 24 h, harvest at 48 h	Decreased: AST, ALT (50–60% reduction), urate, NH_4_ (40% reduction) Reduced steatosis No difference in necrosis	Banas 2009 [28]
ASC, hu	3-stage endoderm/ hepatic	9 days	Up: FOXA2, SOX17, AAT, ALB, ASGR1, HNF4A, TAT, TTR, transferrin, KRT18, GJB1, AFP, 7 CYP genes	Up: ALB, urea synthesis, CYP1A2, CYP 2A1, CYP2E1 activity	CCI_4_ acute NPG mice 2 × 10^6^ cells intrasplenic Harvest survivors on day 8	Survival ALT, AST (magnitude unclear) ALB (rat) increased IF for ALB, AAT at day 8	Xu 2015 [27]
ASC, hu	2-stage endoderm/ hepatic	13 days	Up: AAT, AFP, ALB, AGT, PROS1 Down: KRT18 no chg.: HGF	Glycogen storage, LDL uptake, urea synthesis, HGF secretion, AAT secretion Co-culture with macrophages reduces inflammatory cytokines	CCI_4_ acute, IP C57bl mice 0.5–8 × 10^6^ cells IP, tail vein, intrasplenic Transplant at 6 h post CCI_4_ Harvest at 18 h post-transplant	Improved: AST, ALT (30% reduction), GSH Inflammatory cytokines reduced	Schuur 2021 [45]
Liver MSCs	None	N/A	On: CD73, CD90, CD105 (20%), ALB, vim, ACTA2 Off: KRT18	Glycogen storage Negative for trilineage differentiation	Rag2^−/−^, IL2Rγ^−/−^ male mice Partial hepatectomy 1 × 10^6^ LSC intrasplenic Harvest at 1 or 7 days	Engraftment AST, ALT, bilirubin Regeneration rate (liver weight) Ki67-positive cells	Herrero 2017 [38]

Seven of the 16 studies identified with clinically relevant results used the CCI_4_ model and reported the engraftment of cells, either in the liver or under the kidney capsule [22,26,36,38,39,40,42]. Of these, six of seven studies reported an improvement in clinical parameters, including reduction in total bilirubin, reductions in serum ammonia or blood urea nitrogen (BUN), and reduced aspartate aminotransferase (AST)/alanine amino transferase (ALT) levels [22,26,27,28,43,44]. AST/ALT levels were reduced by approximately 40% compared to untreated controls in these experiments. Ammonia was reduced by approximately 35% in one study, while BUN was reduced by an average of 35%. Total bilirubin was reduced by approximately 35% in the two studies in which it was reported [43,44].

Of the ten studies using models other than CCI_4_, nine described the engraftment of cells [19,35,36,38,39,40,41,42,50]. Five of nine studies reported an improvement in clinical parameters, including reductions in serum ammonia or BUN and reduced AST/ALT levels [35,38,42,45]. AST/ALT levels were reduced by approximately 35% compared to untreated controls in these experiments. Ammonia was reduced by more than 50%, while BUN was reduced by 30%. Two studies reported reductions in steatosis and fibrosis [28,42]. One study reported the stimulation of liver regeneration by mHep treatment [38]. Together, these 16 studies in a variety of liver disease models demonstrate repeated positive impact on parameters generally regarded as important assessments of liver disease. Importantly, three studies (one in the murine CCI4 model, one in the murine galactosamine / lipopolysaccharide [GalN/LPS] model, and one in the rat GalN model) reported improved survival in the mHep treatment group compared to the untreated or MSC control groups [27,35,50].

Several additional clinically important assessments that were tested included inflammation, steatosis, and fibrosis. As noted above, inflammation is one of the key drivers of liver disease. Schuur et al., using mHeps produced from ASCs, were able to demonstrate a reduction in the acute inflammatory cytokines IL-1 beta, IL-6, and TNF alpha in liver tissue in the murine CCI4 model at 24 h post-injury, suggesting that mHeps, such as MSCs, can mitigate acute inflammation [45]. Aurich et al., using the same model and mHeps produced using a similar protocol, were able to show a reduction in steatosis in the livers of treated mice, suggesting a cytoprotective function of mHeps [41]. A similar reduction in steatosis was reported by Bruno et al. in a methionine-choline-deficient diet murine model of non-alcoholic steatohepatitis (NASH) [42]. The cytoprotective function of mHeps was also supported by the maintenance of glutathione levels in the livers of treated mice compared to controls [45]. Downstream sequelae of inflammation in the liver include fibrosis. Bruno et al. were able to also show reductions in fibrosis, as well as reductions in TGF beta, a key cytokine involved in fibrosis, as well as reductions in collagen and smooth muscle actin expression, which is also suggestive of reduced fibrogenic processes [42]. These results were supported by reductions in TGF beta plasma levels observed early after mHep administration in pigs [46].

Finally, an important finding that could be clinically useful is the secretion of elevated levels of cytokines that can assist in the recovery of the liver after acute or chronic injury. Winkler et al. surveyed the cytokine secretome of bone marrow MSCs and mHeps derived from those cells, revealing that mHeps secrete increased numbers and concentrations of cytokines that may be beneficial in liver disease [51]. Schuur et al. also observed an increased secretion of several potentially beneficial cytokines, including stromal-derived factor-1 and other cytokines [45]. Increased HGF secretion was also reported by Bruno et al. [42]. Additional research will be needed to evaluate how altered levels of cytokines that are important in inflammation, fibrosis, and tissue regeneration may impact recovery from disease and injury.

### 2.4. Clinical Experience with mHeps

There is an emerging body of evidence based on clinical applications in humans with liver disease to support the use of mHeps as a therapy in multiple forms of liver disease (Table 3). Allogeneic mHeps from liver MSCs saw early application in treating a 3.5-year-old patient with arginosuccinate lyase (ASL) deficiency [52]. Cells from three donors were administered 11 times by intraportal infusion. A 50% improvement in ammonia levels, a normalization of ASL activity in liver biopsies, and an improvement in psychomotor evaluations were all observed. A second case of ornithine transcarbamylase deficiency in a 3-year-old patient was also treated by an intraportal infusion of mHeps from liver MSCs, again with 11 doses from multiple donors [53]. Ammonia level changes in plasma were suggestive of improvement, and a 3–5% engraftment of cells was detected. Treatment in both of the preceding cases included immunosuppression with tacrolimus. A Phase I study of pediatric hyperammonemia in three patients used the direct injection of mHeps from liver MSCs without immunosuppression [54]. No treatment-related adverse events (AEs) were observed, and the disease was stabilized in all three patients. A second Phase I/II study of mHeps in pediatric patients with urea cycle disorders or Crigler–Najjar syndrome was reported by Smets et al. [55]. In this study, escalating doses of mHeps were administered via the portal circulation. The results presented confirmed the safety and tolerability of mHep administration in this population. Only low levels of antibodies to donor-cell HLA were observed. Ureagenesis was measured and appeared to be elevated in several patients at 3- and 6-months post treatment. Results from a Phase II study in adult patients with acute on chronic (ACLF) liver failure or acute decompensation in patients with underlying chronic liver disease was recently published [56]. mHeps from liver MSCs were infused once or twice into 24 patients using an intravenous (IV) route of administration. Treatment-related AEs (bleeding at the jugular administration puncture site and persistent epistaxis) were observed in the first two patients and were resolved by reductions in the cell dose. Markers of systemic inflammation and of liver function improved over the course of the study. Survival rates in this seriously ill cohort were 83% at 28 days and 71% at 3 months. No surviving patients experienced ACLF as defined by European Association for the Study of Chronic Liver Failure (EASL-Clif) criteria at the 3-month time point.

### 2.5. Manufacturing mHEPs for Use in Clinical Trials

The studies described above provide data that support using mHep cell therapy to treat liver failure. However, a translation of these findings to a product that is clinically useful requires additional significant development. Particularly challenging for cell therapies is scaling production to commercial levels while retaining consistent quality and implementing them at an acceptable price point [32,34,37,57].

Most of the studies described above used laboratory-scale processes to produce cells. Furthermore, the in vivo studies were performed in rodents in all but five of the studies, which may limit the generalizability of the results. Because studies using rodent models require only a few million cells, the consistency of cell production is less of an issue than when large animal or human studies are performed. For such therapies to be tested in the clinic and ultimately to be practical for widespread use in patients, additional process developments are required to both increase the scale of production, define quality metrics, and improve the consistency of production against those quality metrics.

Five of the studies in Table 2 have moved their processes toward the commercial scale. Nickel et al. scaled up the production of mHeps from porcine bone marrow MSCs sufficiently to dose adult pigs (25–30 kg) with 1 × 10^8^ cells, which is in the range anticipated for human doses [46]. Quality control parameters measured include Western blot analyses of livers for phosphoenolpyruvate carboxykinase 1, glutamine synthase, cytochrome P450 1A1, and cytochrome P450 3A4, as well as glycogen storage by PAS stain. In the Phase I study results from Spada et al. on neonates with hyperammonemia, they reported taking production a step further, providing detailed production processes used to generate clinical-quality mHeps for injection [54]. Important points in their description include a cell banking strategy and process control criteria. The potency assay used for Phase I was urea synthesis in vitro after 96 h. Sokal et al. described three clinical applications in which doses of cells for each infusion ranged from 6 × 10^5^ cells/kg to 3.5 × 10^8^ cell/kg in two to 11 infusions. Cell expansion was accomplished using multi-layer cell culture flasks under conventional adherent cell growth conditions [53].

While these studies prove that the scaling of mHep production is feasible for human use at least for the small number of patients required for a phase I study, current methods may not be adequate for commercial-scale production. To offer therapies to patient populations numbering in the thousands and using some simple assumptions about patient numbers (e.g., 10,000) and effective doses, (e.g., 1 × 10^9^), commercial production may need to exceed 2 × 10^13^ cells annually.

Scalability to this level can most easily be achieved using suspension culture systems. A significant body of literature exists on the scalable expansion of MSC-type cells in stirred-tank bioreactor formats [33,57]. Several groups have shown that forming aggregates of MSCs, or embryoid bodies, can not only facilitate scalability, but can also improve in vitro differentiation [19,43,44,50,58]. Of the studies described above that incorporate suspension cultures, the methods described by Xu and Peltz represent the best balance of speed of production (less than 13 days total) and simplicity (two media) [19]. The Xu and Peltz methods have been adapted to liter-scale suspension cultures that are compatible with commercially available stirred-tank bioreactor systems, giving this approach a strong advantage relative to methods using adherent cells or other suspension methods published to date (Hepatx unpublished data) [19].

## 3. Discussion

### 3.1. Both Acute and Chronic Liver Failure Would Benefit from a Multi-Modal Therapy

Because the liver is central to physiological homeostasis, liver dysfunction impacts virtually every organ system. Classic Mendelian diseases, such as sickle cell anemia, can be corrected by replacing the single defective gene, which resolves the hemoglobin gelation that causes the pathophysiology of the disease [59]. In contrast, the loss of hepatocyte function impacts many physiological functions such as ammonia detoxification, other toxin and drug detoxification, hormone metabolism, and clotting factor synthesis. Furthermore, because the liver functions as a protective organ from infectious agents with its high responsiveness to danger signals, it is also primed to respond with inflammation quickly. While this is important for limiting threats to organismal survival, this amplifies the inflammatory response that can lead to additional tissue damage and sensitizes hepatocytes to further death signals [8,10,60]. In the context of ongoing insults, the outcome can be a substantial or complete loss of organ function, which can cascade to multiple organ dysfunction syndrome and death. Clinical best practices rely on treating each of the symptoms individually, for example, by administering steroids for certain types of liver inflammation or fresh frozen plasma or cryoprecipitate for clotting deficiencies, or anticipating surgical bleeding. The effectiveness of liver transplant for liver failure suggests that cell therapy may be able to normalize several aspects of pathophysiology not addressable by individual drug therapies.

### 3.2. There Is Evidence That Liver Failure Can Be Addressed by mHep Therapy

A key feature of cell therapies is that they have the potential to perform multiple cellular functions that promote clinical benefit. MSCs in particular have been demonstrated to deliver multiple functions, including the secretion of trophic and anti-inflammatory factors, as well as novel functions, such as the donation of mitochondria to host cells and the secretion of microRNAs and exosomes that can modulate cell function [61]. Importantly, for liver failure, mHep-type cells can suppress the acute inflammatory signaling from macrophages that is an important driver of hepatitis and progression to late-stage disease [13,15]. MSC-type cells also have the capability to provide trophic support for endogenous cells, including hepatocytes, promoting survival and stimulating proliferative processes involved in regeneration and repair [62]. Each of these functions has been demonstrated in studies using mHeps to treat experimental models of liver disease.

### 3.3. Areas of Strength in mHep Therapy

We have focused on cell therapies for liver injury and liver failure using mHeps. These cells are not identical to fully differentiated hepatocytes but do possess multiple hepatocyte characteristics and functions that can increase their use relative to MSCs in vivo, as detailed above. An important advantage of mHeps from MSCs versus iHeps derived from induced pluripotent stem cells (iPSCs) is that the mHeps retain the anti-inflammatory and pro-regenerative functions of MSCs, in addition to the hepatocyte functions that they express [41,42,44,45]. Although a limited number of studies have been published evaluating the immunomodulatory properties of iPSCs (e.g., Schnabel et al. [63]), none has been published evaluating these properties in iHeps. Therefore, while it is clear that iHeps can perform hepatocyte functions, there is currently no evidence that they are immunomodulatory.

The inflammation observed in liver diseases including alcoholic hepatitis and acetaminophen overdose includes the activation of acute responses with IL-1 beta, IL-6, and TNF alpha [8,9,10,60,64]. These elements of acute inflammation are actively suppressed by mHeps [42,45]. These same inflammatory mediators from macrophages serve to stimulate additional mechanisms of liver disease, including the sensitization of hepatocytes to further damage and the stimulation of hepatic stellate cells to a fibrotic phenotype, and to stimulate infiltration of the liver by inflammatory cells from bone marrow via systemic circulation [8,9,10,60,65,66].

The multiple mechanisms of action of mHeps not only suppress cytokine secretion, but also support the replacement of lost hepatocytes and hepatocyte function: the mHeps provide a cytoprotective function to endogenous hepatocytes, stimulate the proliferation of hepatocytes to replace lost hepatocytes, and can perform some functions normally performed by endogenous hepatocytes (Table 2, cf., Xu [19], Herrera [36], Aurich [40]).

Finally, mHeps can function to reduce fibrosis through anti-fibrotic effects on stellate cells and the increased secretion of fibrinolytic proteins [21]. In fact, one advantage for differentiating MSCs to mHeps compared to using MSCs for liver disease applications is that the liver-specific pro-regenerative functions of mHeps are substantially greater than those of MSCs [45]. This is expected to result in an enhanced clinical benefit relative to MSCs to justify the additional manufacturing steps involved in mHep production.

In addition to the advantages described above, mHeps are capable of engrafting in vivo (Table 2, cf., Aurich [40], Xu [19], Herrero [38]). Additionally, mHeps appear to be able to further differentiate toward fully functioning hepatocytes following engraftment [19,48,67]. Therefore, hepatocyte replacement may be a viable mechanism of action for mHeps, although in acute liver disease models, mHep activity may reflect mechanisms of action that do not require engraftment [45,68]. Although levels of engraftment observed are currently low, the degree of engraftment needed for clinical use is not yet known, and clinically useful levels of hepatocyte replacement may be achievable with less-than-complete liver replacement [16]. Research is currently in progress to improve engraftment of MSC-derived mHeps.

### 3.4. Knowledge Gaps and Alternative Strategies

Although considerable evidence has been published on the efficacy of mHeps in various forms of liver injury and liver disease in animal models, as well as in humans, significant gaps remain in our understanding of how mHeps work. The cells can replace some hepatocyte functions in animal models; however, the precise nature of the impact on disease physiology has not been detailed. Similarly, mHeps can modulate the immune response by macrophages both in vitro and in vivo in animal models, as evidenced by changes in cytokine levels, including IL-1 beta, TNF alpha, and TGF beta [42,45,68]. However, the details of how they mediate this effect are only beginning to emerge. There are hints of additional pro-regenerative activities of mHeps, but few details on the underlying mechanisms are available [68].

MSC-derived exosomes are membrane-bound vesicles that bud from the cells and contain a variety of bioactive molecules, including mRNA, miRNA, and cytokines [69]. An extensive body of literature has been generated characterizing MSC-derived exosomes and their biological activity. MSC-derived exosomes have been demonstrated to mediate immunomodulatory and trophic effects in liver disease models via the various biomolecular cargos they carry [70,71]. Because of these properties, mHep-derived exosomes may represent an alternative way to achieve the therapeutic actions of MSC in a way that is potentially more scalable than whole cells. However, the composition and activity of exosome preparations can vary dramatically depending on the culture conditions under which they are produced [69,70,71]. Therefore, additional research is needed to facilitate the clinical translation of exosomes for liver disease.

A large number of studies have been published that evaluate native MSCs, or so-called “primed” MSCs, in liver disease (see Yang et al. [72] and Noronha et al. [73] and the references therein). Native MSCs from multiple sources (most often bone marrow, adipose tissue, and umbilical cord) have significant immunomodulatory and pro-regenerative effects. Although several studies have been published comparing MSCs from various sources with respect to these properties, no consensus has emerged regarding a preferred source for application in liver disease. Furthermore, these beneficial properties of MSCs can, under certain circumstances, be enhanced by “priming” with interferon gamma (reviewed in Noronha et al. [73]). While this may be advantageous, some studies suggest that, in the context of liver disease, the mHep approach may provide increased immunomodulatory function as well as liver-specific functions such as ureagenesis and drug detoxification by the cytochrome P450 system (See Bogliotti 2022 [68], Aurich 2008 [41], and Bruno 2019 [41]).

### 3.5. Future Directions

To fulfill the clinical promise of mHeps, two things must be completed. First, the production of mHeps must be standardized. As with all cell therapies, resolving these production issues can take substantial time and resources. The differentiation protocols presented in Table 2 illustrate the variability in the currently used production method, which, along with the variability in the donor source of the MSC, produces a high level of variability in the characteristics of the mHeps. Further work is needed to define the critical quality attributes that are important for the clinical success of mHeps.

Second, careful clinical development will be required to understand how to optimally use mHeps in a clinical setting. Based on the properties of our current mHeps and liver disease pathology described earlier, we suggest that patients with acute inflammatory liver disease may benefit from this therapy. Since it is likely that a subset of this diverse group of patients may be most likely to respond, a better assessment of the patient characteristics which contribute to a successful outcome is needed. In addition, issues including optimal dose, dose timing, route of administration, and excipients may make the difference between success and failure in clinical development. If carefully developed, mHeps may be able to deliver a clinical benefit for these patients where conventional drug therapy, including biologics, have not. Beyond helping these patients who currently have little hope, knowledge gained from the development of this new type of therapy can form a foundation for the development of additional ground-breaking cell therapies and may help to shift thinking about how to treat patients with complex diseases.

## Figures and Tables

**Figure 1 cells-11-01998-f001:**
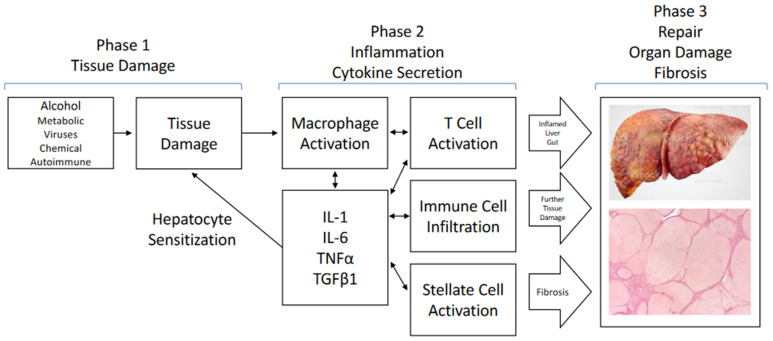
Common pathways in liver injury and disease. Tissue damage, whether from either disease processes or chemical injuries, initiates complex inflammatory responses involving several liver cell types, typified by stellate cells, T cells, as well as macrophages/Kupffer cells. Secretion of multiple cytokines and chemokines amplifies the inflammatory cascade, further sensitizing hepatocytes to damage, and leads to fibrosis. These damaging processes often extend beyond the liver to include gut, pancreatic, and kidney damage, with secondary effects on cardiac function. Chronic activation of these processes ultimately leads to aberrant tissue healing and scarring.

**Table 1 cells-11-01998-t001:** Evolution of the proocols used for differentiating MSC-type cells to mHeps.

Number of Stages	Key Components per Stage	Key Characteristics	Advantages	Disadvantages	Reference
1	Stage 1: HGF, FGF, DMSO, OSM	Express KRT18, ALB, AFP, HNF1 alpha, GATA4, FOXA2 Glycogen storage, urea synthesis	Simplicity	Length of differentiation	Schwartz 2002 [20], Seo 2019 [21] 2005 [22]
2	Stage 1: FGF Stage 2: HGF, ITS, dexamethasone	Express KRT18, ALB, AFP, HNF1 alpha, HNF-3b Glycogen storage, urea synthesis	Efficiency of differentiation	Complexity	Snykers 2006 [23]
2	Stage 1: HGF, ITS, FGF Stage 2: OSM/LIF, ITS, dexamethasone	Express ALB, TDO2, AAT, TAT, CK8, CK19, AFP, CX32, G6P Glycogen storage, urea synthesis	Simplicity	Length of differentiation	Lysy 2008 [24]
3	Stage 1: EGF, FGF Stage 2: HGF, FGF, ITS, nicotinamide Stage 3: OSM, nicotinamide, dexamethasone, ITS	Express ALB, TDO, AAT, AFP, CNX32 Glycogen storage, urea synthesis	Efficiency of differentiation	Length of differentiation	Campard 2008 [25]
3	Stage 1: WNT pathway activator, FGF Stage 2: HGF, FGF, OSM, ITS, dexamethasone, nicotinamide Stage 3: Nicotinamide, dexamethasone	Express ALB, TOD2, FOXA2, Sox17, AAT, ALB, ASGR1, HNF4A, TAT, TTR, transferrin, KRT18, GJB1, AFP, 7 CYP genes Glycogen storage, urea synthesis, CYP activity, albumin secretion	Rapid differentiation, efficiency of differentiation	High complexity	2007 [26], Xu 2015 [27], Banas 2009 [28]
3	Stage 1: WNT pathway activator, FGF, activin Stage 2: HGF, FGF, OSM, ITS, dexamethasone	Express ALB, TOD2, FOXA2, SOX17, AAT, ALB, ASGR1, HNF4A, KRT18, AFP Glycogen storage, urea synthesis, albumin secretion	Rapid differentiation, efficiency of differentiation, scalability	Moderate complexity	Xu 2014 [19]

**Table 3 cells-11-01998-t003:** Clinical studies using mHeps.

Cell Type	Study Design	Outcomes	Reference
Liver MSCs	Hu Phase I clinical study: 3 pts.: inherited neonatal hyperammonemia Dose level 1: 1.25 × 10^5^ cells per gm liver (pt. 1) Dose level 2: 2.5 × 10^5^ cells per gm liver (pts. 2 and 3) ROA direct injection into liver parenchyma No immunosuppression	No treatment-related AEs- Stable disease until transplant No immune response to cells	Spada 2019 [54]
Liver MSCs	Hu case report 3.5 y.o. female Argininosuccinate lyase deficiency 11 cell infusions with male cells Portal vein ROA 1 × 10^9^ cells per dose	Ammonia levels improved (50% reduction) Psychomotor evaluation improved Cytogenetics on biopsies (12% hepatocyte replacement at 12 months) ASL activity in biopsies increased to normal levels	Stéphenne 2006 [52]
Liver MSCs	Hu case report 3 y.o. female Ornithine transcarbamylase deficiency 11 cell infusions with cells from multiple donors Portal vein ROA ~2 × 10^8^–7 × 10^8^ cells per dose	Engraftment: 3–5% Ammonia level changes suggestive of improvement	Sokal 2013 [53]
Liver MSCs	Hu Phase I/II Hu Pediatric patients with urea cycle disorders and Crigler–Najjar syndrome	Safety and tolerability of treatment confirmed Evidence of improvements in ureagenesis observed	Smets 2019 [55]
Liver MSCs	Hu Phase II clinical study ACLF and AD 24 pts. 6 × 10^5^–5 × 10^6^ cells/kg IV ROA	Safety: no serious treatment-related AEs; other AEs as expected for this patient population. First 2 pts. had bleeding issues, so dose lowered for remaining pts. Systemic inflammation improved in group over study Liver function improved in group over study	Nevens 2021 [56]

## Data Availability

Not applicable.

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
