# Peer review of "Clinical Application of Induced Hepatocyte-like Cells Produced from Mesenchymal Stromal Cells: A Literature Review"

_cells, 2022, doi:10.3390/cells11131998_

Round 1

Reviewer 1 Report

In this paper, Bogliotti Y et al reviewed recent results of studies employing novel cell therapies using hepatocyte-like cells manufactured from mesenchymal stromal cells to treat liver disease. The authors first described the complex pathophysiology of liver disease and evidenced common pathways in liver injury and disease. They explain the different protocols used for differentiating MSC-type cells to hepatocytes and interestingly described the advantages and disadvantages of these differentiation methods. They reported the effects of MSC-derived mHeps on liver disease models. Clinical studies using mHeps were also well analyzed and described. The manufacturing of mHeps for use in clinical trials was debated. Indeed the production of mHeps must be standardized. The differentiation protocols are very variable as well as the MSC source. The authors suggest that cell replacement therapy can be safe and effective in liver patients for whom there is no other option.

Overall the manuscript is well organized and well written, and I think it will be helpful for readers of Cells. This is a very extensive review and discussion of the literature concerning novel cell therapies using mHeps to treat liver disease. Tables are clear and well structured.

The authors can consider following suggestions to improve the review:

Table 1: The protocol used by Sokal E et al shoulb also described (Differentiation 2008, Gastroenterology 2008). They differentiated BM- or UC-MSC using a cocktail of HGF, FGF4, Oncostatin and DXM for a 20 day maturation.

Table 3 : the studies on animal models and human clinical studies should be separated.

MSC immunomodulatory properties can be stimulated to enhance their basal anti-inflammatory molecular potential. Ex vivo priming protocols have reported the ability to modulate and/or boost MSC therapeutic properties. The authors should also comment how preconditioning of MSC (hypoxia, cytokines, TLR challenging) can influence the therapeutic potential of mHeps derived from MSC focusing on immunomodulatory and regenerative features.

Concerning strength areas of mHep therapy, the authors suggest an important advantage of mHeps from MSC versus iHeps derived from induced pluripotent stem cells (iPSCs) in terms of anti-inflammatory and pro-regenerative functions. The authors should better explain  these differences with appropriate references.

Concerning the future directions, the authors should also discuss about the potential regenerative medicine using synergistic mixtures of immature and mature hepatocytes with MSC and hepatic stellate cells.

Author Response

We appreciate the thoughtful comments from both reviewers.  Below we have added those comments, as well as responses to those comments indented and in blue.

Reviewer 1: Comments and Suggestions for Authors

In this paper, Bogliotti Y et al reviewed recent results of studies employing novel cell therapies using hepatocyte-like cells manufactured from mesenchymal stromal cells to treat liver disease. The authors first described the complex pathophysiology of liver disease and evidenced common pathways in liver injury and disease. They explain the different protocols used for differentiating MSC-type cells to hepatocytes and interestingly described the advantages and disadvantages of these differentiation methods. They reported the effects of MSC-derived mHeps on liver disease models. Clinical studies using mHeps were also well analyzed and described. The manufacturing of mHeps for use in clinical trials was debated. Indeed the production of mHeps must be standardized. The differentiation protocols are very variable as well as the MSC source. The authors suggest that cell replacement therapy can be safe and effective in liver patients for whom there is no other option.

Overall the manuscript is well organized and well written, and I think it will be helpful for readers of Cells. This is a very extensive review and discussion of the literature concerning novel cell therapies using mHeps to treat liver disease. Tables are clear and well structured.

The authors can consider following suggestions to improve the review:

Table 1: The protocol used by Sokal E et al shoulb also described (Differentiation 2008, Gastroenterology 2008). They differentiated BM- or UC-MSC using a cocktail of HGF, FGF4, Oncostatin and DXM for a 20 day maturation.

Indeed, the protocols from the Sokal group should be added to the appropriate tables.  To that end, new 2 and 3 stage protocols added to table 1. Minor edits were made to the associated text to reflect the diversity of protocols for mHep preparation.  Three new references were added to match the newly added papers.

Table 3: the studies on animal models and human clinical studies should be separated.

We agree with this change.  The animal and human studies are now in separate tables.

MSC immunomodulatory properties can be stimulated to enhance their basal anti-inflammatory molecular potential. Ex vivo priming protocols have reported the ability to modulate and/or boost MSC therapeutic properties. The authors should also comment how preconditioning of MSC (hypoxia, cytokines, TLR challenging) can influence the therapeutic potential of mHeps derived from MSC focusing on immunomodulatory and regenerative features.

We considered that MSC licensing protocols (eg. Using IFNg and TNFa) could also increase the immunomodulatory capacities of the final mHEPS giving them a higher immunomodulatory potential in vivo, in addition to their pro-regenerative and hepatic functional properties.

However, we observed a naturally occurring potentiation of immunomodulatory effects in our mHEPs versus their starting ASCs (Bogliotti ISCT 2022) without licensing as a consequence of the differentiation protocol. Therefore, we view differentiation of MSC-type cells into the hepatocyte pathway (i.e., mHeps) as a type of licensing or preconditioning in a therapeutic sense, one that adds features to the cells that are useful in the context of liver disease and will be superior to traditional MSC licensing.  Initial results support this, however, additional studies are needed to confirm this hypothesis.

We have added text in section 3.4 on p.19 that discusses the above concepts.

Concerning strength areas of mHep therapy, the authors suggest an important advantage of mHeps from MSC versus iHeps derived from induced pluripotent stem cells (iPSCs) in terms of anti-inflammatory and pro-regenerative functions. The authors should better explain these differences with appropriate references.

There are few published studies that have evaluated the immunomodulatory properties of iPSC prior to differentiation and none that we are aware of that have examined these properties in iHeps.  The text in section 3.3 on p.18 has been modified to reflect this.

Concerning the future directions, the authors should also discuss about the potential regenerative medicine using synergistic mixtures of immature and mature hepatocytes with MSC and hepatic stellate cells.

Although we agree that the above mixtures of cells are interesting and may have therapeutic potential, we believe that the technology is far from clinical application.  Our focus in this review is on the practical clinical application of mHeps in liver disease, seeking to emphasize that the technology is at or near clinical stage.  The focus of research in these mixtures of cells, which include organoids, has been biology of differentiation for the most part.  Very little is known about their regenerative or immunomodulatory potential.  In addition, a substantial hurdle to clinical application and commercialization is manufacturing such mixtures of cells.  For these reasons we feel that trying to fairly present this approach as a future direction relevant to mHeps is not a great fit.

Reviewer 2 Report

There are a large number of review articles about mesenchymal stromal cells (MSCs). While the topic is not new, the review is systematic.This review focuses on the clinical application of hepatocyte-like cells derived from MSCs. There are a few minor concerns:

1. MSCs could be derived from different sources, which whether might affect the clinical application of hepatocyte-like cells derived from MSCs.

2. Compared with the direct use of MSCs therapy,what are the differences, advantages and disadvantages between MSCs and mHeps for clinical treatment?

Author Response

We appreciate the thoughtful comments from reviewers. Below we have added those comments, as well as responses to those comments indented and in blue.

Reviewer 2: Comments and Suggestions for Authors

There are a large number of review articles about mesenchymal stromal cells (MSCs). While the topic is not new, the review is systematic. This review focuses on the clinical application of hepatocyte-like cells derived from MSCs. There are a few minor concerns:

  1. MSCs could be derived from different sources, which whether might affect the clinical application of hepatocyte-like cells derived from MSCs.

As we describe in the review, mHeps can be made from MSCs derived from multiple sources.  In terms of preclinical properties, there is no clear difference between the sources based on the published literature.  Although reports have been published comparing MSCs from different sources for their immunomodulatory properties, similar reports have not been published for mHeps.  This would be an interesting subject for future developmental studies, but at this point there is no suggestion in the literature that one source is better than another.  To clarify these ideas we have added text on p.6 to present these conclusions.

  1. Compared with the direct use of MSCs therapy, what are the differences, advantages and disadvantages between MSCs and mHeps for clinical treatment?

Unfortunately, in the literature the primary comparison of the starting MSCs and mHeps derived from them has been efficiency of engraftment in vivo.  In that regard, mHeps are clearly superior.  mHeps also obviously possess hepatocyte functional characteristics that the starting MSCs don’t have, such as ureagenesis.  In addition, we have observed that immunomodulatory properties of the cells are increased in mHeps compared to their starting MSCs (Bogliotti ISCT 2022).

At the clinical level no comparison has been made between MSCs and mHeps.  There is a whole literature on treatment of liver disease with MSCs, which is beyond the scope of this review as we conceived of it as is discussed in section 3.4 on p.19.